# Infectious Complications Following Kidney Transplantation—A Focus on Hepatitis C Infection, Cytomegalovirus Infection and Novel Developments in the Gut Microbiota

**DOI:** 10.3390/medicina55100672

**Published:** 2019-10-04

**Authors:** Samuel Chan, Nicole M Isbel, Carmel M Hawley, Scott B Campbell, Katrina L Campbell, Mark Morrison, Ross S Francis, E Geoffrey Playford, David W Johnson

**Affiliations:** 1Department of Nephrology, Princess Alexandra Hospital, Brisbane, QLD 4102, Australia; nikky.isbel@health.qld.gov.au (N.M.I.); carmel.hawley@health.qld.gov.au (C.M.H.); scott.campbell@health.qld.gov.au (S.B.C.); ross.francis@health.qld.gov.au (R.S.F.); david.johnson2@health.qld.gov.au (D.W.J.); 2Australasian Kidney Trials Network, The University of Queensland, Brisbane, QLD 4102, Australia; katrina.campbell@health.qld.gov.au (K.L.C.); geoffrey.playford@health.qld.gov.au (E.G.P.); 3Translational Research Institute, Brisbane, QLD 4102, Australia; 4Centre for Applied Health Economics, Menzies Research Institute, Griffith University, Brisbane, QLD 4102, Australia; 5The University of Queensland Diamantina Institute, Faculty of Medicine, University of Queensland, Woolloongabba, QLD 4102, Australia; m.morrison1@uq.edu.au; 6Infection Management Services, Department of Microbiology, Princess Alexandra Hospital, Brisbane, QLD 4102, Australia

**Keywords:** cytomegalovirus, direct acting antivirals, donor-derived infections, gastrointestinal microbiome, hepatitis C, knowledge acquisition, letermovir, kidney transplantation

## Abstract

The incidence of infectious complications, compared with the general population and the pre-transplant status of the recipient, increases substantially following kidney transplantation, causing significant morbidity and mortality. The potent immunosuppressive therapy given to prevent graft rejection in kidney transplant recipients results in an increased susceptibility to a wide range of opportunistic infections including bacterial, viral and fungal infections. Over the last five years, several advances have occurred that may have changed the burden of infectious complications in kidney transplant recipients. Due to the availability of direct-acting antivirals to manage donor-derived hepatitis C infection, this has opened the way for donors with hepatitis C infection to be considered in the donation process. In addition, there have been the development of medications targeting the growing burden of resistant cytomegalovirus, as well as the discovery of the potentially important role of the gastrointestinal microbiota in the pathogenesis of post-transplant infection. In this narrative review, we will discuss these three advances and their potential implications for clinical practice.

## 1. Introduction

Kidney transplant recipients have a greatly increased risk of infection-related morbidity and mortality compared with the general population and the pre-transplant status of the recipient [1]. Worldwide, the incidence of infectious complications following kidney transplantation has been reported to range between 49 to 80% [1]. This increased risk is likely due to various immunosuppressive medications required to prevent allograft rejection [1,2]. Donor-transmitted infections, such as Hepatitis C virus (HCV), may also contribute to a heightened risk of infectious complications post-transplantation and have generally resulted in organs from high risk donors (e.g., HCV-positive) being excluded from transplant consideration. In addition to the virus itself, there is emerging data that HCV is associated with increased incidence of bacteremia, ventilator associated pneumonia and catheter-related bloodstream infection [3]. New treatments for donor-transmitted infections, such as direct acting antiviral drugs for HCV, substantially mitigate this risk [4]. Furthermore, cytomegalovirus (CMV) resistance appears to be emerging, which has necessitated the development of newer generation treatments to counteract this problem [5]. Finally, there is emerging evidence that kidney transplant recipients may have significantly altered gastrointestinal microbiota, which in turn may be associated with increased risks of infection as a result of transmural migration of bowel micro-organisms, altered immunosuppressive medication pharmacokinetics and progressive kidney disease [6,7,8]. This review will discuss these three recent, key, promising, innovative approaches to potentially mitigating infectious burden in kidney transplant recipients through direct acting antiviral drugs targeting HCV, new treatments for CMV resistance, and therapeutic manipulation of the gut microbiota.

## 2. Changes in the Management to Donor-Derived Infections

Donor-derived infections in kidney transplant recipients causes significant morbidity and mortality [9,10]. The incidence of donor-derived infections has been reported to be approximately 0.2% in solid organ transplantation [11]. Donor-derived infections may be classified into either expected or unexpected infections [12]. Expected donor-derived infections, namely CMV and Epstein-Barr Virus, may be identified by donor-recipient screening. Unexpected donor-derived infections are always a potential concern and cannot be completely excluded, and may include viral and bacterial infections such as urinary tract infections [10,11]. Of all the donor-derived infections reported, viral infections are most frequent [9,10].

Hepatitis C virus (HCV) is a well-recognized donor-derived infection. HCV positive donors were previously not offered to HCV uninfected recipients because of the increased mortality associated with liver and cardiovascular diseases [13,14]. Since 1995, 3502 HCV seropositive kidneys have been discarded in the United States of America, which is equivalent to a rate of 53.6% compared with 22.4% of HCV negative kidneys [15,16]. HCV seropositive but nucleic acid testing (NAT) negative donors are increasing in number and have been increasingly accepted as donors, with the caveat that a window-period is considered if the donor has had persistent risk behavior [17]. These donors also represent patients who have been successfully treated for HCV [17]. Recipients of these IgG positive NAT negative kidneys have been shown to become HCV IgG-positive post-transplant but not NAT-positive [18,19]. The likely explanation for this observation appears to be the transfer of HCV peptide with the organ or the transfer of passenger anti-HCV antibodies producing lymphocytes in the graft [20]. Nonetheless, the management of donor-derived HCV infection has been revolutionized through the development of direct-acting anti-HCV (DAA) drugs. HCV positive donors are a large pool, are usually younger compared with donors without HCV infection, and have fewer co-morbidities that are more likely to increase recipient and organ survival [13,14,21,22]. Recipients of HCV RNA positive donors or high risk donors (e.g., history of intravenous drug use, incarceration, less safe sexual practices, etc.,), particularly within the NAT window period, should have ongoing post-transplant surveillance for the appearance of HCV infection [23]. However, the exact timing of the surveillance is still being refined [23].

There are three main classes of DAAs which are classified based on the specific proteins that are targeted on the HCV [4,5,6,7,8,9,10,11,12,13,14,15,16,17,18,19,20,21,22,23,24] (Table 1). The protease inhibitors which act on the NS3 part of the HCV RNA include boceprevir, telaprevir, simeprevir, sunaprevir, grazoprevir and paritaprevir. The NS5A inhibitors, acting on the NS5A part of the HCV RNA, include daclatasvir, ledipasvir, ombitasvir, elbasvir and valpatasvir. The polymerase inhibitors, acting on the NS5B part of HCV RNA, include sofosbuvir and dasbuvir. These agents are orally administered, and the treatment duration varies between eight and twenty-four weeks [25,26]. Genotype I has the broadest DAA treatment options; however, all other genotypes have at least one DAA option. Suitable direct-acting antiviral agent combination regimens for each HCV genotype are depicted in Table 2.

Compared with traditional HCV treatments such as interferon alpha and ribavirin, DAAs have emerged as a highly effective therapeutic option for managing kidney transplant recipients who receive HCV RNA positive donor kidneys [4,5,6,7,8,9,10,11,12,13,14,15,16,17,18,19,20,21,22,23,24]. Interferon alpha is contraindicated in kidney transplant recipients because of the significant risk of provoking acute rejection [27]. In a study of 7344 adult patients with chronic HCV infection enrolled from 32 hepatology centers in France who were followed up for approximately 33 months, it was found that DAAs were associated with lower mortality (adjusted hazards ratio (HR) 0.48, 95% CI 0.33–0.70) and hepatocellular carcinoma (adjusted HR 0.66, 95% CI 0.46–0.93) rates and were not associated with decompensated cirrhosis (adjusted HR 1.14, 95% CI 0.57–2.27) [28]. A Cochrane systematic review which included 138 randomized clinical trials and 25,232 participants, concluded that DAAs were relatively expensive and there was insufficient evidence to suggest that DAAs reduced mortality or other liver-related complications from chronic HCV [29]. However, this review has been heavily criticized for its interpretation and conclusions given its methodological flaws and the overall lack of clinical insight and knowledge of the natural history of HCV [29].

Three studies have examined the efficacy and tolerability of utilizing HCV-positive donors (HCV RNA positive and NAT positive) into appropriately consented HCV-negative kidney transplant recipients (Table 3). The first study [30] was the Transplanting Hepatitis C Kidneys into Negative Kidney Recipients (THINKER) trial. In this open-label, singe-group pilot trial, 10 HCV negative patients received kidneys from HCV positive donors (9 had genotype 1a infection), were administered elbasvir-grazoprevir from day 3 following transplantation, and were cured of HCV which was defined as a sustained virologic response 12 weeks after transplantation. The mean 6-month estimated glomerular filtration rate was 62.8 mL/min/1.73 m^2^. The 12-month follow-up data on these 10 recipients, as well as 6-month follow-up data on additional 10 HCV-negative recipients of HCV-positive kidneys demonstrated that all 20 recipients had undetectable HCV RNA from 4 weeks until the end of follow-up of 6 months [31]. The THINKER participants had similar kidney graft function at 12 months compared to recipients of HCV-negative kidneys who met THINKER criteria and were matched for donor Kidney Donor Profile Index score (median 72.8 vs 67.2 mL/min/1.73 m^2^). Notably, one HCV-negative recipient developed subnephrotic range proteinuria without kidney function impairment and 5 recipients experienced transient serum aminotransferase elevations [31].

The third study was the Exploring Renal Transplants Using Hepatitis C Infected Donors for HCV Negative Recipients (EXPANDER) trial [32]. In this single center, single arm, open-label, non-randomized study, kidneys from 10 HCV-positive donors (genotypes 1–3) were transplanted into 10 HCV-negative recipients aged over 50 years and then treated for 12 weeks with grazoprevir, elbasvir and, for genotype 2 or 3 infections, sofosbuvir. At the end of the treatment period, no participants had detectable HCV RNA or treatment-related adverse drug reactions. Table 3 also illustrates studies [33,34,35,36] in which HCV RNA positive non-kidney solid organs were transplanted into HCV-negative recipients.

Although these studies add to the evidence supporting the transplantation of HCV-positive kidneys in HCV-negative kidney recipients with subsequent treatment with DAAs, it should be noted that they were industry sponsored and limited by small sample sizes, single center study designs and short follow-up durations, which reduce the certainty of the evidence. A number of studies evaluating the efficacy, safety and tolerability of DAAs in seronegative patients of HCV positive donors are currently underway [37,38,39]. Furthermore, treatment protocols are still being developed and refined at the current time of writing this manuscript—the exact timing of introducing DAAs, the optimal duration of treatment, the appropriate monitoring requirements—are important questions and will require ongoing investigator-initiated multicenter studies to evaluate this.

A major consideration in using DAAs in kidney transplant recipients has been the optimal time to treat HCV infection. Problems related to limited drug access have emerged when using DAAs [43]. Severe membranoproliferative glomerulonephritis requiring haemodialysis, for example, has been reported in a 61-year old HCV-naïve diabetic male who received a liver transplant when glecaprevir/ pibrentasvir was unavailable until 24 days following transplantation [33].

Another important consideration in using DAAs is the high wholesale costs ranging from US $417 (glecaprevir/pibrentasvir) to US $1125 (ledipasvir-sofosbuvir) per day [43]. A study which used data from the Canada Health System created a Markov model to examine the cost-effectiveness of utilizing deceased HCV donors for kidney transplantation in HCV-negative recipients [44]. This study showed a cost-effectiveness ratio of $56,018 per quality adjusted life years (QALY) from the payer perspective and $4647 per QALY from the societal perspective, compared with recipients who would otherwise have remained on dialysis for an additional year [44].

Another issue emerging may be the potential risks of inducing resistance associated with transplanting different genotypes in kidney transplant recipients receiving DAAs. Resistance may occur particularly when therapeutics levels are suboptimal, thus creating selective pressure for resistant HCV to emerge as the dominant species [45]. A study undertaken in Brazil involving 76 patients, of whom 39 were kidney transplant recipients and 37 were on chronic haemodialysis, examined the prevalence of resistance-associated substitutions to DAAs and found that the overall prevalence of resistance was 38.2% with substitution resistance detected in NS3A (17.8%), NS5A (21.9%) and NS5B (8.4%) inhibitors [46]. Resistance substitutions were higher in Genotype 1a (42.9%) compared with Genotype 1b (32.4%) (*p* = 0.35) [46]. However, this study was limited by the fact that patients were restricted to Genotype 1 and the sample size was small. More studies will be required to further elucidate the resistance patterns of DAAs.

In addition to being effective in the treatment of recipients of kidneys from HCV-positive donors, DAAs appear also to be effective for the treatment of HCV-positive recipients. A retrospective Italian study a sustained virologic response in 12 (92%) of 13 HCV RNA-positive kidney transplant recipients [47]. There is ongoing debate regarding whether it is better to treat HCV-positive individuals with end-stage kidney disease before or after a kidney transplant [48]. Early treatment prior to kidney transplantation may reduce the risks of hepatic complications, dialysis transmission of HCV, post-transplant glomerulonephritis and post-transplant diabetes mellitus, whilst treatment following kidney transplantation affords the patient the opportunity to receive a kidney from a HCV-positive donor thereby shortening transplant waiting time [48]. Our practice is to treat HCV-infected individuals as soon as possible prior to kidney transplantation.

## 3. New Approaches to the Management of Infections in the Era of Antimicrobial Resistance

A paradigm of antimicrobial resistance developing in kidney transplant recipients involves cytomegalovirus (CMV), which is an opportunistic viral pathogen causing infection and disease with significant morbidity and mortality. Indeed, 60% of kidney transplant recipients will have an active CMV viraemia, and more than 20% will develop symptomatic disease [49,50,51,52]. Infection with CMV usually develops when prophylaxis is ceased and may cause end-organ damage such as hepatitis, pancreatitis or pneumonitis [50,51]. Four antiviral therapies are currently marketed for either the prophylaxis and/or treatment of CMV infection: ganciclovir, the ganciclovir prodrug (valganciclovir), foscarnet and cidofovir.

According to current guidelines, options for CMV prophylaxis include oral valgancyclovir, oral valaciclovir, and intravenous ganciclovir [53]. The addition of anti-CMV immunoglobulin to these agents has not been shown to have any additional benefit. Although valganciclovir is used most frequently in many kidney transplant units because of its oral formulation, it is limited by high costs and occasional difficulties with access. Intravenous ganciclovir, on the other hand, is cheaper and more readily available but limited due to the difficulties in giving it in the home environment [53]. The recommended dosage for CMV prophylaxis is 900 mg for oral valgancyclovir daily and 3200 mg for oral valaciclovir daily for 3 months in CMV seropositive recipients, adjusted for kidney function [53]. Some kidney transplant units have used a lower dose of valganciclovir for CMV prophylaxis which may in turn lead to resistance; however, more studies will be required to assess the efficacy and potential resistance patterns of valgancyclovir at a lower dose [53]. For kidney transplants involving CMV seromismatch (i.e., donor seropositive, recipient seronegative), a duration of 6 months is recommended. The alternative strategy to prophylaxis for prevention of CMV disease is routine viral load monitoring and prescribing antiviral treatment when viral loads increase significantly regardless of whether or not the individual is symptomatic (pre-emptive treatment). Whilst the Updated International Consensus Guidelines on the Management of Cytomegalovirus in Solid-Organ Transplantation indicate that there is moderate evidence supporting this approach [53], a previous Cochrane review of the efficacy of pre-emptive therapy compared to prophylaxis concluded that the evidence was uncertain due to the presence of appreciable study heterogeneity [53,54]. Monitoring of viral loads for up to 6 months following CMV prophylaxis in patients with established risk factors for CMV should occur [53].

Mutations in UL-97 and UL-54 mediate CMV resistance to the above therapies [53,55,56,57]. The incidence of CMV resistance varies between 2% to 7% [51,52]. Risk factors include CMV donor positive/recipient negative serostatus, potent immunosuppressive use, induction therapy with anti-thymocyte globulin, high viral loads and prolonged duration of treatment with suboptimal drug levels [51,52]. A few different antiviral therapies, such as letermovir and maribavir, are currently being studied to mitigate CMV resistance [53,55,56,57,58]. The pharmacology of these two therapies are summarized in Table 4. The adoptive transfer of autologous or third-party CMV-reactive T-cells is also being examined as a potential therapy.

Letermovir is a new non-nucleoside CMV inhibitor which targets the viral terminase complex [59,60], and has been demonstrated to inhibit CMV in both in vitro and in vivo preclinical studies [60,61]. In a phase 3, double-blind, randomized controlled trial of letermovir versus placebo in 565 CMV-seropositive adult hematopoietic-cell transplant recipients, 495 participants had undetectable levels of CMV DNA at baseline. Amongst these individuals, the occurrence of clinically significant CMV infection by 24 weeks was significantly lower in those receiving letermovir compared with placebo (38% vs. 61%, respectively, *p* <0.001). Myelotoxic and nephrotoxic adverse effects were comparable in both groups [62]. Similar findings have been reported in kidney transplant recipients. In a multi-center, open-label, randomized controlled trial of letermovir (40 mg twice a day or 80 mg once a day) or usual care in 27 kidney transplant recipients with active CMV replication, viral clearance was more often achieved in the combined letermovir groups (6 out of 12, 50%) than the usual care group (2 out of 7, 29%) [62]. There were no reported relapses in CMV during this trial [62].

A case report by Kau et al. reported the first successful treatment of multidrug-resistant CMV with letermovir in a 39-year-old male lung transplant recipient who developed severe CMV pneumonitis, retinitis and colitis that was refractory to ganciclovir, foscarnet, CMV hyperimmune globulin, cidofovir, artemether/lumefantrine and leflunomide [63]. CMV genotype analysis demonstrated A594T and C603W UL97 mutations. Following introduction of letermovir in combination with immunosuppression reduction, the patient made a rapid recovery.

On the contrary, there has been some recent literature regarding potential resistance in letermovir. A case series reported 2 heart and 2 lung transplant adult patients who received letermovir for treatment of ganciclovir-resistant CMV disease after failing therapy with ganciclovir and valganciclovir, and developing nephrotoxicity from foscarnet [64]. Two of the 4 patients from this study had CMV retinitis proven by CMV PCR obtained via anterior chamber paracentesis and the other 2 patients had funduscopic examination consistent with retinitis. Induction letermovir doses occurred at 720 mg and in one patient titrated to 960 mg and all 4 patients had clinical improvement in retinitis [64]. However, 3 of the 4 patients failed to achieve sustained virologic suppression, raising potential concerns for letermovir resistance. Genotypic assessment demonstrated UL56 mutations. Virologic suppression occurred in the 3 patients when transitioned back to original therapy. However, none of the patients reported adverse effects secondary to letermovir [64]. Overall, there is some evidence that letermovir may show promise as a treatment for CMV resistance. However, the role of letermovir in patients with CMV disease who cannot tolerate currently available therapies remains to be determined. Moreover, the dosing, efficacy and potential resistance patterns of letermovir in the management of CMV resistance requires further evaluation.

In addition to letermovir, maribavir, an inhibitor of UL-97 viral kinase, is also currently under clinical trials for managing CMV resistance, although results are still in the preliminary phase [65]. Papanicoloaou et al. [65] conducted a randomized, double-blinded, dose-ranging phase 2 study of 3 different doses of maribavir (400 mg, 800 mg or 1200 mg twice daily) for up to 24 weeks in 47 hematopoietic-cell and 73 solid-organ transplant recipients (including 30 kidney transplant recipients) with active CMV infection that was refractory or resistant to ganciclovir, valganciclovir, foscarnet and cidofovir (defined as failure to achieve at least a 1 log10 decrease in CMV DNA viral load after at least 2 weeks of treatment). Overall, 67% (95% CI 57–57%) of patients achieved undetectable CMV DNA within 6 weeks of maribavir treatment, with similar results observed in the 3 dosage groups (400 mg 70%, 800 mg 63%, 1200 mg 68%). Balanced against these benefits, 68% developed adverse events, including dysgeusia (65%) and neutropenia (11%), which led to maribavir discontinuation in 34% of patients. Recurrent CMV infections occurred in 25% and 4 patients (3.3%) died due to CMV.

Another important investigation conducted by a French group included 12 patients (3 bone marrow recipients and 9 solid organ transplant recipients) with CMV resistance, showed that half the patients responded to maribavir when trialed at 800 mg daily doses [66]. Although maribavir shows promise in the treatment of resistant CMV infections, concerns regarding maribavir resistance have emerged [65] with T409M and H411Y being reported as potential gene mutation markers [67]. Other studies have raised concerns regarding the efficacy of maribavir, with one randomized, double-blinded, multicenter controlled trial of 303 liver transplant recipients showing that maribavir 100 mg twice daily was ineffective at preventing CMV infections [68]. Collectively, these findings highlight that further studies are required to determine the efficacy, tolerability and potential for resistance in using maribavir for the management of resistant CMV infections in kidney transplant recipients.

An alternative approach to the management of CMV resistance has been the use of adoptive T-cell therapy with CMV-reactive T-cells [69,70,71]. CMV control is critically dependent on effective T cell immunity. In a prospective, multicenter, single-arm, open-label, non-randomized phase 1 study [71] of in vitro–expanded autologous CMV-specific T cell therapy in 13 solid organ transplant recipients with recurrent or ganciclovir-resistant CMV infection, 11 (84%) displayed either complete resolution or reduction in DNAemia. It has also been demonstrated that multiple infusions may be required, particularly if the initial response was suboptimal or if rebound CMV viraemia occurred [69,70]. Other groups have developed multivirus reactive T cell protocols typically including viruses such as CMV, Epstein-Barr virus, BK virus and adenovirus [72,73,74]. Difficulties inherent with such treatments include high cost, the appreciable time taken to adequately generate the T-cells for transfusion, and the requirement for significant patient commitment to adhere to hospital appointments for treatment success [69,70]. The potential benefits of adoptive T-cell therapy also need to be balanced against the potential associated risks of treatment. For example, graft failure (*n* = 1, 2%) and graft-associated thrombotic microangiopathy (*n* = 1, 2%) have been reported in a group of 50 allogenic stem cell transplant patients who were given a single dose of CMV-specific T-cells [75]. Given the limited sample size in the above studies, more research is required to assess the robustness of using autologous T-cell therapy in CMV resistance.

## 4. Emergence of Gastrointestinal Microbiota and Transplant Associated Infections

In recent times, it has been recognized that a major, potential source of infection in immunocompromised individuals is the gut microbiota, which is comprised of bacteria, archaea, fungi, protozoa, and their respective viruses [76,77,78]. A large number of observational and cross-sectional studies have shown that the gut microbiota is a functional and dynamic interface linked with immune regulation, metabolic modulation, food digestion, angiogenesis promotion, gut epithelial health, energy homeostasis, neurobehavioural development and drug absorption, metabolism and disposition [79,80,81]. Additionally, antibiotic use, psychosocial and physical stress, radiation, dietary changes and various disease states are all known to be associated with alterations in the taxonomic and functional properties of the gut microbiota [82]. These changes are generically referred to as “dysbiosis”. In the setting of gut dysbiosis, transmural migration of gut micro-organisms and/or their toxic products (endotoxins and uraemic toxins such as indoxyl sulphate and p-cresyl sulphate) may lead to infection, inflammation, endotoxaemia, and the progression of kidney disease [83,84,85,86]. Products of the gut microbiota (e.g., peptidoglycans, polysaccharide A) also interact with the enteric immune system to stimulate both innate and adaptive immune mechanisms, and antigen cross-reactivity may promote alloimmunity and rejection through molecular mimicry [87].

Several studies have demonstrated that the gut microbiota may be significantly altered in the setting of kidney transplantation and play an important role in post-transplant outcomes. Gut microbiota in kidney transplant recipients can be potentially modified by immunosuppression, antibiotic administration, dietary changes, altered bowel mobility and even transplantation of microbiota via kidney and kidney-pancreas allografts [82,88,89]. In a pilot study in which microbiota profiles were examined in serial fecal specimens from 26 kidney transplant recipients during the first 3 months post-transplant using polymerase chain reaction (PCR) amplification of the 16S rRNA V4-V5 variable region, Lee et al. demonstrated significant changes in gut microbiota profiles compared to pre-transplantation [6]. Importantly, higher fecal abundance of Enterococcus was associated with Enterococcus urinary tract infection and pre-dated the occurrence of this infection by up to 39 days. Median fecal abundance of Enterococcus in transplant recipients who did and did not develop Enterococcus urinary tract infection was 24% and 0%, respectively (*p* = 0.005). The group also demonstrated that acute rejection was associated with higher fecal abundance of *Enterococcus, Clostridium tertium, Anaerofilum and* Lactobacillales, and lower fecal abundance of *Bacteroides*, *Ruminococcus,* Lachnospiraceae, Clostridiales, *Blautia, and Eubacterium dolichum.* Another group also reported associations between fecal abundance of micro-organisms and the occurrence of infection and acute rejection in kidney transplant recipients [90]. Lee et al. subsequently reported that post-transplant diarrhea was associated with reductions in fecal diversity measures and specifically, decreases in the relative abundance of commensal bacteria such as Ruminococcus, Dorea, Coprococcus and Bacteroides spp., rather than with common infectious diarrheal pathogens [7]. Finally, another study of serial fecal specimens in 19 kidney transplant recipients during the first post-transplant month has demonstrated that alterations in gut microbiota profiles might be associated with altered immunosuppressant medication pharmacokinetics [8]. Specifically, the relative abundance of one bacterium, Faecalibacterium prausnitzii, was significantly greater in kidney transplant recipients who ultimately also required at least a 50% increase in tacrolimus dosing over the first month to achieve a target serum level of 8–10 ng/mL, when compared to recipients who did not require such a dosage escalation (11.8% versus 0.8%, respectively, *p* = 0.002) [8]. Taken together, these preliminary findings involving relatively small patient numbers suggest that kidney transplantation results in significant changes in the gut microbiota composition which in turn are associated with important surrogate and clinical post-transplant outcomes including infection, altered serum immunosuppressant medication levels, acute rejection and post-transplant diarrhea.

Manipulation of the gut microbiota through nutritional interventions, such as prebiotics, probiotics, and synbiotics, may therefore represent a novel approach to mitigating infection and other transplant complications such as rejection and post-transplant diarrhea [91,92]. Although there have been no studies that have specifically addressed nutritional interventions targeting the gut microbiota in kidney transplant recipients, a meta-analysis of four studies (3 randomized controlled trials and 1 historically controlled trial) involving 246 liver transplant recipients has shown that administration of prebiotics and probiotics resulted in appreciably reduced rates of overall infection (relative risk (RR) 0.21, 95% CI 0.11–0.41, I2 1%), urinary tract infection (RR 0.14, 95% CI 0.04–0.47, I2 0%) and intra-abdominal infection (RR 0.27, 95% CI 0.09–0.78, I2 0%) [93]. The interpretative strength of this review is limited by the small number of available studies, the heterogeneity of the prebiotic and probiotic interventions, small patient numbers, short follow-up durations, inclusion of a non-randomized controlled trial and low certainty of the evidence. Currently, the safety of probiotics in the kidney transplant population is uncertain, and their use is therefore not routinely recommended at the present time [94].

Recently, there has been evidence emerging that fecal microbiota transplantation (FMT) may be an effective option for manipulating the gut microbiota, particularly in the setting of recurrent *Clostridium difficile* infection. In a retrospective singe-center chart review of 35 patients with recurrent *Clostridium difficile* who underwent FMT in the United States of America, 85.7% (*n* = 30) reported resolution of symptoms approximately 6 to 8 weeks post-transplant [95]. Adverse effects were monitored by the research team, but none were reported in the final study [95]. Eight of the 35 recipients were reportedly receiving immunosuppressive therapy, although the type and dosage of such therapy was not specified [95]. A subsequent systematic review and meta-analysis of 54 non-randomized studies of FMT in 303 immunocompromised patients with recurrent *Clostridium difficile* infection reported success rates of 87% on first treatment [96]. FMT has also been evaluated as a potential therapy for steroid-resistant acute graft-versus-host disease in the setting of stem cell transplantation [97] FMT was safely tolerated and effective in 4 patients, with 3 experiencing a complete response and one having a partial response. Although these results appear to be promising, the role of FMT in manipulation of the gut microbiota in kidney transplant recipients has yet to be determined.

Indeed, examining the role(s) of the gut microbiota in the pathogenesis of infections in transplant recipients present considerable challenges. There will always be significant heterogeneity amongst transplant recipients, attributable to their previous medical history including hospital admissions, prior exposure to various antibiotics prior to transplantation, as well as immunosuppressive therapy and antimicrobial prophylaxis. However, the findings from the small number of studies to date do suggest that adequately powered, well-designed, and multi-center randomized controlled trials are justified to determine whether and how variations in the gut microbiota can be translated into low cost, prognostic and/or therapeutic approaches that reduce post-transplant infections, as well as to maximize the use of immunosuppressive therapy post-transplantation (e.g., tacrolimus).

## 5. Conclusions

Over the last five years, ongoing research has led to significant advancements in the field of kidney transplant infectious disease medicine. The development of new approaches to manage donor-derived infections, such as Hepatitis C, have allowed expansion of the deceased kidney donor pool to include donors that were previously considered unsuitable. At the same time, new antiviral agents, such as letermovir and maribavir, are currently being trialed to combat growing CMV resistance. Recent evidence has also suggested that the gut microbiota, which changes appreciably following kidney transplantation, might represent a significant source of post-transplant infections, and contribute to altered immunosuppressive agent pharmacokinetics, acute rejection and post-transplant diarrhea. These outcomes may be mitigated by nutritional interventions (e.g., pre, pro- and synbiotics) and fecal microbiota transplantation, although further studies are required to comprehensively evaluate their safety and efficacy. These developments have generated considerable research interest and endeavor in the transplant infectious disease field and offer new opportunities to alleviate infectious morbidity and mortality in kidney transplant patients.

## Figures and Tables

**Table 1 medicina-55-00672-t001:** Pharmacology of direct-acting antivirals agents.

Agent Class	Example	Genotype	Adverse Events	Drug-Drug Interactions	Contraindications	Probability of Drug Resistance
NS3/4A protease inhibitors	Boceprevir Telaprevir Simeprevir Asunaprevir Paritaprevir Grazoprevir	Narrow	Fatigue Anemia Nausea Dysgeusia Headache	Multiple via CYP3A and p-glycoprotein (e.g., ritonavir, erythromycin, rifampicin, efavirenz)	Low creatinine clearance; use of alpha-1 adrenoreceptor antagonists, anticonvulsants, oral contraceptive pills	High
NS5A inhibitors	Daclatasvir Ledipasvir Ombitasvir Elbasvir Velpatasvir	Medium	Headache Fatigue Nausea Diarrhea Insomnia	Minimal; case reports of thyroid hormone, dihydropyridines, alpha and beta blockers, proton pump inhibitors, statins	Low creatinine clearance; previous Hepatitis B, use of systemic steroids and anticonvulsant therapy	Low
NS5B polymerase inhibitors	Sofosbuvir (nucleoside) Dasabuvir (non-nucleoside)	Broad (nucleoside) Narrow (non-nucleoside)	Fatigue Symptomatic bradyarrhythmias	Minimal	Low creatinine clearance; use of anticonvulsant and antimicrobial therapy, HIV protease inhibitor therapy and herbal supplements (e.g., St John’s Wort)	Low

**Table 2 medicina-55-00672-t002:** Suitable direct-acting antiviral agent combination regimens for each Hepatitis C virus (HCV) genotype.

Genotype	Suitable Regimens
Genotype 1	Ledipasvir-sofosbuvir Sofosbuvir-velpatasvir Elbasvir-grazoprevir Glecaprevir-pibrentasvir Dasabuvir-omitasvir-paritaprevir-ritonavir Ombitasvir-paritaprevir-ritonavir-daclatasvir
Genotype 2	Sofosbuvir-velpatasvir Glecaprevir-pibrentsvir Dalatasvir-sofosbuvir Sofosbuvir-ribavirin
Genotype 3	Glecaprevir-pibrentasvir Sofosbuvir-velpatasvir Dalatasvir-sofosbuvir Sofosbuvir-ribavrin
Genotype 4	Ledipasvir-sofosbuvir Sofosbuvir-velpatasvir Elbsvir-grazoprevir Glecaprevir-pibretasvir
Genotype 5 and 6	Ledipasvir-sofosbuvir Sofosbuvir-velpatasvir Glecaprevir-pibrentasvir

**Table 3 medicina-55-00672-t003:** Studies evaluating Hepatitis C positive donors into Hepatitis C negative recipients in transplantation.

Study	Year	Study Design	N	Mean Age of Recipients (Years)	Organ Transplant	Intervention	Results
Durand [31]	2018	Open-label, non-randomised trial	10	71 (median)	Kidney	Grazoprevir/elbasvir (Genotype 1); sofosbuvir added for Genotype 3	HCV RNA not detectable Graft function stable No treatment adverse effects (sponsor: Merck Pharmaceuticals)
Reese [30]	2018	Open-label, non-randomised trial	20	56	Kidney	Grazoprevir/elbasvir (Genotype 1)	HCV RNA not detectable Graft function stable No treatment adverse effects (sponsor: Merck Pharmaceuticals)
Woolley [32]	2019	Open-label, non-randomised trial	44	61 (median)	Heart and lung	4 week-regimen of sofosbuvir/velpatasvir	35/44 enrolled and completed 6 months follow-up (non-detectable HCV RNA, stable graft function no treatment adverse effects)
Wadei [33]	2019	Case report	1	-	Liver	Glecaprevir/pibentasvir (delayed)	Delay in direct-acting anti-HCV drugs (DAA) resulted in severe membranoproliferative glomerulonephritis requiring haemodialysis
Abdelbasit [34]	2018	Case series	5	47	Lung	Sofosbuvir/ledipasvir (Genotype 1); sofosbuvir/alpatasvir (Genotype 2)	HCV RNA not detectable Graft function stable No treatment adverse effects
Schlendorf [35]	2018	Open-label, non-randomised trial	13	53	Heart	Ledipasvir/sofosbuvir (Genotype 1) and Sofosbuvir/Velpatasivr (Genotype 3)	12/13 undetectable HCV RNA 1 death (pulmonary emboli) Graft function stable No treatment adverse events
Cotter [40]	2019	Registry	2635	57	Liver	Various DAAs (registry study)	3-year graft survival following use of DAAs increased to 88% from 79%
Axelrod [41,42]	2018	Registry	157,873 kidney and 58,509 liver transplants	-	Kidney and liver	Various DAAs (registry study)	Improvements in graft function and death post DAA
Cholankeril [41,42]	2018	Registry	3855	58	Liver	Various DAAs (registry study)	1-year post transplant survival pre-DAA 89.9% vs 91.9% post DAA

**Table 4 medicina-55-00672-t004:** Pharmacology of letermovir and maribavir.

Medication	Mechanism of Action	Adverse Events	Resistance Patterns	Drug-Drug Interactions
Letermovir	Inhibits viral terminase complex (UL51/JL56/UL89)	Nausea, diarrhea, vomiting, peripheral edema, cough, headache, fatigue and abdominal pain	None noted	None known, possibly with ciclosporin
Maribavir	Inhibits CM UL97 serine/threonine kinase by competitively inhibiting the binding of ATP to the kinase ATP-binding site	Gastrointestinal disorders (diarrhea, dysgeusia, nausea, vomiting)	Emerging (T409M and H411Y)	CYP3A4 P-glycoprotein

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
