# Peer review of "Infectious Complications Following Kidney Transplantation—A Focus on Hepatitis C Infection, Cytomegalovirus Infection and Novel Developments in the Gut Microbiota"

_medicina, 2019, doi:10.3390/medicina55100672_

Round 1

Reviewer 1 Report

General Comments :

This is an interesting review regarding recent and important advances in the field. The relative scarcity of authors interpretation makes the review difficult to read. Some paragraphs are very long and just enumerate the study’s findings, the main message is very difficult to extract for the reader. The link between HCV and microbiota is unclear if any. The gut microbiota paragraph is almost off topic. HIV and PHS high risk donors are not discussed. Major revisions are warranted.

Specific comment :

Page 1: Authors say “New treatments for donor-transmitted infections, such as direct acting antiviral drugs for HCV, may substantially mitigate this risk and allow expansion of the donor pool”. Why “may”? Transplanting HCV+ kidney to HCV+ or HCV- patients is currently performed in several centres and “is” increasing the donor pool. Same thing with HIV, however within study protocols.

Page 1: Authors say “Finally, there is emerging evidence that kidney transplant recipients may have significantly altered gastrointestinal microbiota”. Assessing gut microbiota is interesting but not part of routine clinical practice. What is the direct relation with kidney transplant? What is the clinical message behind this sentence? Just providing a reference is not so helpful, digest of the main conclusions would be appreciated.

Page 2: “infection3”. Please correct.

Page 2: “DAAs are more efficacious and tolerable”. Please expand, why, how?

Re: “HCV donors for seronegative patients” It would be nice to mention the currently ongoing studies. Cf. clinicaltrial.gov

Table 1: Please add the low creatinine clearance as a contraindication. Please add recommended treatment start after surgery, duration. Adding the genotype in the footnote is not helpful because hard to read, please find another way of showing that.

“?substrate » What does that mean ?

Table 2: is there really only one case report transplanting HCV+ liver? E.g. Cotter at al. “Increasing Utilization and Excellent Initial Outcomes Following Liver Transplant of Hepatitis C Virus (HCV)‐Viremic Donors Into HCV‐Negative Recipients: Outcomes Following Liver Transplant of HCV‐Viremic Donors”

Can the authors describe the current clinical practice with respect to CMV prophylaxis? What dose, how long, in which case? Also describe the alternative, CMV monitoring, please describe the recommended frequency overtime.

Page 5 : “Another important study was that of Kau et al” studying one case does not represent “study” per se.

Page 7: “Over the past five years new evidence…” please provide references.

Page 7: “4. Emergence of gastrointestinal microbiota and transplant associated infections”. Please descibe the clinical implications of these findings; impact on immunosuppressive drug levels? Etc..

Page 7: please note that probiotic may be unsafe in transplant population, their use is currently not recommended.

Some transplanted patients with C diff were treated with fecal transplant. This subject could be discussed in more details. E.g. C. Caenepeel et al. Faecal microbiota transplantation as treatment for recurrent clostridium difficile infections: a single center experience

HIV is not discussed, of note HIV+ to HIV+ liver and kidney transplant with different genotype were performed. Re HCV and HIV, a discussion about the potential risks of transplanting different genotype and induce resistance should be discussed in more details. This should also be done for HCV.

The transplantation of “Public Health Service (PHS) high risk donors” is very frequent in the US. I would recommend addressing this interesting topic. Transplanting organs from these donors requires specific testing (e.g. HIV, HCV, HBV etc) after transplant.

Author Response

Dear Sir/ Madam,

Thank you very much for your helpful comments. Please see the attachment for a response to each of the comments, and the corresponding manuscript with tracked changes, highlighting the alterations made.

Yours sincerely
Samuel Chan

Reviewer 2 Report

In this interesting review entitled  Recent advances in management of infectious complications following kidney transplantation, the authors discuss new approaches in understanding and combating infectious complications of kidney transplantation, including new treatments for donor-transmitted HCV infection, new treatments for CMV resistance, and new knowledge regarding the gastrointestinal microbiota.

The paper must be globally reviewed concerning English grammar, because is really difficult to follow. In addition, in my opinion there are some criticism that make this paper not suitable in this form for publication in Medicina Journal.

Major recommendations

-          First of all, I recommend, if the authors want maintain the settings of this review in this way, to change the title. For example: Infectious complications following kidney transplantation: focus on HCV, CMV and novel knowledge in microbiota.  However, I think that those are 3 different topics, that are not really suitable to be discussed in this way, so the authors must explain clearly in the introduction the reason why they structured the review in this way. 

-          In the introduction, some concepts about the epidemiology of the infectious risk and prevalence during the time of transplantation should be done.

Fishman JA. Infection in Organ Transplantation. Am J Transplant. 2017;17(4):856-879.

-           

-          A section of the review should be reserved to the importance of the pre-transplant evaluation of the donor and of the recipient. In particular, the analysis of some “dialysis parameters” should be reported.

-          HCV section:

o   Please spend some words also about the HCV DAAs treatments of the already HCV+ patients (before transplantation).  

Gendia M, Lampertico P, Alfieri CM, D'Ambrosio R, Gandolfo MT, Campise MR, Fabrizi F, Messa P. Impact of hepatitis C virus and direct acting antivirals on kidney recipients: a retrospective study. Transpl Int. 2018 Dec 23.

-       CMV section:

o   Dedicate some words reporting novel evidences about Pre-transplant factors related to CMV activation after renal transplantation. In particular, stress the problem of the immunosuppressive therapy (especially induction therapy!!)

o   Explain the different protocols of CMV prevention also accordin the recent evidences.

The Third International Consensus Guidelines on the Management of Cytomegalovirus in Solid-organ Transplantation. Kotton CN, Kumar D, Caliendo AM, Huprikar S, Chou S, Danziger-Isakov L, Humar A; The Transplantation Society International CMV Consensus Group. Transplantation. 2018 Jun;102(6):900-931

o   Explain better the importance and the future clinical utility of Quantiferon test.

-           

 Minor recommendations 

In HCV section there is a mistake: “infection3” instead than “infection”.

References 41-44 in the text should be indicated in apex.

Author Response

(The authors gave the same response as above.)

Round 2

Reviewer 1 Report

General Comments :

The authors responded to my comment, although partially in some respects.  

Specific comment :

An additional thing that the authors could mention is the importance of assessing the microbial flora that an organ is bringing with itself and possibly transmitted to the donor when transplanted. This is particularly relevant with kidney-pancreas transplant, where the donor pancreas duodenum flora is transplanted together with both organs. Of note, an in depth study of organ preservation media has an impact on endocrine function (Meier RPH et al. Pancreas preservation fluid microbial contamination is associated with poor islet isolation outcomes – a multi-centre cohort study, Transplant International 2018). The kidney alone, also come with its own flora in some occasions (Sharma AK et al. Clinical outcome of cadaveric renal allografts contaminated before transplantation. Transplant International 2005). The authors could then connect the dots and underline the importance of the importance of the surveillance of the microbiota both in the donor and the recipient.

Author Response

Dear Sir/ Madam,
Thank you very much for your comments. Please see the attachment.
Yours sincerely
Samuel Chan

Reviewer 2 Report

All the topics suggested were assessed. In this form, the paper is more suitable for the publication in Medicina.

Author Response

Dear Sir/ Madam,
Thank you very much for your positive feedback.
Yours sincerely
Samuel Chan